# Reduced Chitosan as a Strategy for Removing Copper Ions from Water

**DOI:** 10.3390/molecules28104110

**Published:** 2023-05-16

**Authors:** Pedro M. C. Matias, Joana F. M. Sousa, Eva F. Bernardino, João P. Vareda, Luisa Durães, Paulo E. Abreu, Jorge M. C. Marques, Dina Murtinho, Artur J. M. Valente

**Affiliations:** 1University of Coimbra, CQC-IMS, Department of Chemistry, 3004-535 Coimbra, Portugal; petermatias1998@gmail.com (P.M.C.M.); uc2011141774@student.uc.pt (J.F.M.S.); eva.fbernardino@gmail.com (E.F.B.); paulo.abreu@ci.uc.pt (P.E.A.); qtmarque@ci.uc.pt (J.M.C.M.); dmurtinho@ci.uc.pt (D.M.); 2University of Coimbra, CIEPQPF, Department of Chemical Engineering, 3030-790 Coimbra, Portugal; jvareda@eq.uc.pt (J.P.V.); luisa@eq.uc.pt (L.D.)

**Keywords:** reduced chitosan, adsorption, copper ions, water purification

## Abstract

Toxic heavy metals are priority pollutants in wastewater, commonly present in dangerous concentrations in many places across the globe. Although in trace quantities copper is a heavy metal essential to human life, in excess it causes various diseases, whereby its removal from wastewater is a necessity. Among several reported materials, chitosan is a highly abundant, non-toxic, low-cost, biodegradable polymer, comprising free hydroxyl and amino groups, that has been directly applied as an adsorbent or chemically modified to increase its performance. Taking this into account, reduced chitosan derivatives (RCDs 1–4) were synthesised by chitosan modification with salicylaldehyde, followed by imine reduction, characterised by RMN, FTIR-ATR, TGA and SEM, and used to adsorb Cu(II) from water. A reduced chitosan (RCD3), with a moderate modification percentage (43%) and a high imine reduction percentage (98%), proved to be more efficient than the remainder RCDs and even chitosan, especially at low concentrations under the best adsorption conditions (pH 4, *R*_S/L_ = 2.5 mg mL^−1^). RCD3 adsorption data were better described by the Langmuir–Freundlich isotherm and the pseudo-second-order kinetic models. The interaction mechanism was assessed by molecular dynamics simulations, showing that RCDs favour Cu(II) capture from water compared to chitosan, due to a greater Cu(II) interaction with the oxygen of the glucosamine ring and the neighbouring hydroxyl groups.

## 1. Introduction

The fight against climate change advocates reducing all forms of pollution and a strong focus on sustainability. Heavy metal pollution is associated with several anthropogenic activities, such as mining, industrial production, and agriculture [1,2]. The practice of these activities is problematic when pollutant discharges into ecosystems become substantial, causing an accumulation of metals in the environment [1] and their biomagnification in the food chain [3]. Furthermore, only a few metals have biological functions, and most of them, including copper, nickel, cadmium, and lead, are highly toxic, causing several illnesses [4,5,6]. Despite these issues, they remain the main pollutants in European groundwater and soils [7] and are found in dangerous concentrations in many places across the globe [1]. Therefore, tackling heavy metal pollution is not only a matter of public health but can also be a way of preserving ecosystems. In this sense, adsorption technologies are currently widely applied as a cost-effective, efficient, versatile, and simple method for the environmental remediation of a wide variety of pollutants, including heavy metals [8,9].

In addition to the combat of pollution, waste reduction is also crucial for sustainable development. Hereupon, circular economy efforts not only reduce waste but also promote the reuse of materials, increasing the sustainability of industrial processes. This recycling adds value to waste and creates new jobs to assist in its collection, separation, and processing. For instance, fishery waste has significantly increased in the last decades, as fish is wasted and discarded in the production and distribution chains, which has economic and environmental impacts. As a result, our society needs better resource management and to use waste biomass for high-added-value applications [10]. Chitin is the second-most-abundant natural polysaccharide and can be extracted from fishery waste, as it is a structural polymer of arthropods’ shells [11]. On the other hand, chitosan is a partially *N*-deacetylated chitin derivative [11] for which the market was estimated to be 107 thousand tonnes in 2020, as this polymer is widely used in antiseptics, food processing, medicine, wastewater treatment, etc., because of its antibacterial and antifungal characteristics, biocompatibility, biodegradability, and ability to chelate metals [10,11,12,13,14].

Due to increasing seafood production, cheap chitosan is widely available and can be upcycled into high-added-value applications [15,16,17]. This work highlights its application in wastewater treatment, particularly in heavy metal sorption from water [12], which is an approach of paramount interest capable of reducing alarming levels of aqueous pollution while using functional material derived from the reuse of waste.

The sorption of heavy metals by chitosan has been addressed by several authors, either by using chitosan with different deacetylation degrees and molecular weights [18,19] or by using chemically modified chitosan [20,21]. Since the performance of pure chitosan for adsorption does not attain the desired level, this paper describes a new, simple, and effective strategy based on the reduction of an imino-modified chitosan for copper(II) removal from water. Salicylaldehyde was selected as an inexpensive natural compound for the modification of chitosan, as it presents an aldehyde group that can react with the chitosan amine groups, as well as an aromatic ring and an additional hydroxyl group that can improve the metal complexation [22]. Consequently, since imine bonds can undergo hydrolysis in water, their reduction ensures the stability of the polymer and restores the amine groups, which are well-known as metal coordination groups and sites that can improve chelation [23,24]. Furthermore, the mechanism of interaction is also discussed and unveiled. To reach this goal, molecular dynamics (MD) simulations have been carried out to obtain detailed insights into the interaction between Cu(II) and chitosan or reduced chitosan derivative.

## 2. Experimental Section

### 2.1. Reagents

Chitosan of low molecular weight (deacetylation degree > 75%; 50,000–190,000 Da; 20–300 cP for 1 wt. % in 1% acetic acid at 25 °C) was purchased from Aldrich (Schnelldorf, Germany). For chitosan modification, salicylaldehyde (99%, Merck, Darmstadt, Germany) and sodium borohydride (98%, Thermo Fisher Scientific, Geel, Belgium) were required. Ethanol (96%, José Manuel Gomes dos Santos, Lisboa, Portugal), methanol (≥99.8%, Riedel-de Haën, Seelze, Germany), nitric acid (65%, PanReac AppliChem, Barcelona, Spain), acetic acid (>99.9%, Sigma-Aldrich, Schnelldorf, Germany), deuterated water (99.9%D, Eurisotop, Saint-Aubin, France) and deuterium chloride (99%D, Eurisotop, Saint-Aubin, France) were used as solvents. In adsorption tests, copper(II) acetate monohydrate (≥98%, J. T. Baker, Center Vally, PA, USA), nickel(II) acetate tetrahydrate (98%, Sigma-Aldrich, Schnelldorf, Germany), lead(II) acetate trihydrate (≥99%, J. T. Baker, Center Vally, PA, USA) and cadmium(II) acetate dihydrate (≥99.8%, Riedel-de Haën, Seelze, Germany) were used. 

### 2.2. Apparatus

A Bruker Avance III, 400 MHz spectrometer was used to acquire proton nuclear magnetic resonance spectroscopy (^1^H-RMN) spectra at room temperature (r.t.); samples for these measurements were prepared by dilution of 7.5 mg of polymers in 1 mL of 1% DCl in D_2_O (*v*/*v*).

An Agilent Technologies Cary 630 FTIR spectrophotometer was utilised for infrared attenuated total reflection spectroscopy (FTIR-ATR) analysis, in the range of 4000–650 cm^−1^.

A Nietzsch Tarsus TG 209 F3 device was utilised to obtain thermogravimetric (TG) profiles between 25 and 800 °C, using a nitrogen flux of 50 mL min^−1^, a heating rate of 10 °C/min, and 2–3 mg of sample placed in an alumina (Al_2_O_3_) crucible.

A Zeiss Merlin Gemini 2 field-emission scanning electron microscope, equipped with an energy-dispersive spectrometer (EDS) capability for chemical composition analysis, was used to record scanning electron microscopy (SEM) images.

An Unicam Solaar 939 spectrometer was employed to carry out flame-atomic absorption spectroscopy (F-AAS) experiments using an air/acetylene flame and a hollow cathode lamp for the direct determination of each heavy metal in aqueous solutions: copper (325 nm), nickel (232 nm), cadmium (229 nm) and lead (217 nm). Standard solutions for calibration curves were prepared using a stock solution of 1000 mg L^−1^ of each metal ion (Cu(II), Ni(II), Cd(II) or Pb(II)) in 0.5 M nitric acid (from PanReac AppliChem, Barcelona, Spain).

### 2.3. Synthesis of Reduced Chitosan Derivatives (RCDs)

Four different chemically modified chitosan derivatives, labelled RCD1 to RCD4, were prepared. For this, chitosan (0.250 g) was dissolved under stirring in 25 mL of the solvent described in step (A) of Table 1, at room temperature (r.t.). Complete dissolution of chitosan was achieved after 90 min. Then, 0.125 mL of salicylaldehyde diluted in 2 mL of methanol was added drop by drop, forming a yellowish viscous solution. This solution was then stirred at 40 °C overnight and a yellow gelatine was obtained. After cooling to r.t., 25 mL of solvent described in step (B) of Table 1 was added and the mixture was stirred at 70 °C under reflux for 1 h, in order to decrease its viscosity. After the solution cooled, sodium borohydride was added in two steps (2 × 0.450 g) spaced two hours apart. The reduction reaction was allowed to proceed under reflux at 70 °C overnight. Finally, the addition of ethanol ensured the complete precipitation of the corresponding polymer and the solid was filtered, washed with ethanol and finally with distilled water until neutralisation. In all cases, the obtained light-yellow product was dried overnight at 40 °C, powdered using a mortar and pestle, and stored in a desiccator until further use [25,26].

The effect of different synthetic strategies on the degree of modification (*DM*) (referring to the first step) and reduction (*DR*) (second step) of RCDs were assessed by ^1^H-NMR spectroscopy (Appendix A) [27], using Equations (1) and (2), respectively,
(1)$$DM=\left(\frac{\frac{A_{H8-11}}{4}}{A_{H2}}\right)\times 100$$
(2)$$DR=\left(1-\frac{\frac{A_{H7}}{A_{H2}}}{\frac{DM}{100}}\right)\times 100$$
where *A_H_*_8–11_ represents the area of the four aromatic protons of the salicylaldehyde ring, *A_H_*_2_ is the area of the proton in C2 of the pyranose ring, and *A_H_*_7_ is the area of the proton in imine function (referring to the non-reduced residual amount). The degree of deacetylation (*DD*) of commercial chitosan was also measured using the following equation:(3)$$DD=\left(1-\frac{\frac{A_{3\times H12}}{3}}{A_{H2}}\right)\times 100$$
where *A_H_*_12_ corresponds to the area of protons of the acetyl group [22,27].

### 2.4. Heavy Metal Adsorption

Quantification of adsorption processes at equilibrium was performed by calculating sorption efficiencies (*Q*, %) (Equation (4)) and the amount of metal ion sorbed per gram of adsorbent (*q_e_*, mg g^−1^) (Equation (5)), knowing the initial (*C*_0_, mg L^−1^) and equilibrium (*C_e_*, mg L^−1^) metal ion concentrations, the volume of solution (*V*, L) and the mass of sorbent (*m*, g).
(4)$$Q\left(\%\right)=\frac{C_{0}-C_{e}}{C_{0}}\times 100$$
(5)$$q_{e}=\frac{\left(C_{0}-C_{e}\right)\times V}{m}$$

The effects of solid–liquid ratio, pH, initial concentration and contact time on Cu(II) adsorption by chitosan and RCDs were evaluated. Copper solutions were prepared by dissolving the aforementioned salt in ultrapure water in order to simulate synthetic wastewater. 

All sorption analyses were performed by using a weighted polymer mass in 5 mL of an aqueous metal solution, and then the sample was shaken at 120 rpm and 25 °C for 24 h in an incubator (ZWI-100H, LABWIT). 

A preliminary test was performed to assess the best chemically modified chitosan to adsorb Cu(II), using a Cu(II) solution of 100 mg L^−1^ at pH = 4 and *R*_S/L_ = 2.0 mg mL^−1^. Subsequent optimisation studies were carried out by changing the solid–liquid ratio (1.0–4.0 mg mL^−1^) and pH (3–6). 

Sorption isotherms and kinetics were carried out by using the best sorption experimental conditions, i.e., *R*_S/L_ = 2.5 mg mL^−1^ and pH = 4. 

To investigate the sorption mechanism, the experimental equilibrium data were best described by Langmuir [28] and Langmuir–Freundlich [18] models (Equations (6) and (7)), respectively:(6)$$q_{e}=\frac{q_{m}K_{L}C_{e}}{1+K_{L}C_{e}}$$
(7)$$q_{e}=\frac{q_{m}{\left(K_{LF}C_{e}\right)}^{b}}{1+{\left(K_{LF}C_{e}\right)}^{b}}$$
where *q_m_* (mg g^−1^) is the maximum adsorption capacity, *K_L_* (L mg^−1^) is the Langmuir constant, *K_LF_* ((L mg^−1^)^1/*b*^) is the Langmuir–Freundlich constant and *b* is the Langmuir–Freundlich heterogeneity constant.

Pseudo-first-order and pseudo-second-order kinetic Equations (8) and (9), respectively) [6] were used to assess the kinetic mechanism of sorption.
(8)$$q_{t}=q_{e}\left(1-e^{-k_{1}t}\right)$$
(9)$$q_{t}=\frac{k_{2}q_{e}^{2}t}{1+k_{2}q_{e}t}$$
where *q_t_* (mg g^−1^) is the amount of metal ion adsorbed at a defined time *t* (min), and *k*_1_ (min^−1^) and *k*_2_ (g mg^−1^ min^−1^) are the pseudo-first- and pseudo-second-order rate constants, respectively. 

The goodness-of-fit for the adsorption isotherms and kinetics models was evaluated through the coefficient of determination (*R*^2^) and the Akaike information criterion (*AIC*). *AIC* measures the prediction error and, therefore, the smaller its value the greater the fitting quality will be. It can be determined by Equation (10), in which *n* is the number of experimental points, *s*^2^ is the residual sum of squares and *K* is the number of model parameters [13].
(10)$$AIC=n\times log\left(\frac{s^{2}}{n}\right)+2K$$

Selectivity tests for the adsorption of Cu(II) were performed with synthetic wastewater containing four different metal ions: Cu(II), Ni(II), Cd(II) and Pb(II), at 4 × 10^−4^ M each. 

Before quantification by F-AAS, the polymer-containing samples were filtered through a Nylon syringe filter with a pore size of 0.22 μm, and all solutions were acidified by adding 5% (*v*/*v*) of nitric acid 0.5 M. To analyse any secondary adsorption to the container surface, several polymer-free control experiments were also performed, and it was demonstrated that no secondary adsorption occurred. In addition, all sorption experiments were carried out at least in duplicate.

### 2.5. Computational Methodology

#### 2.5.1. Systems

In this work, the possibility of adsorbing Cu(II) with chitosan and functionalised chitosan (RCD) in aqueous solutions has been studied. Since the experiments were carried out at an acidic pH, protonation is expected to occur in the nitrogen atoms of the amine groups. Consequently, the structural units of protonated chitosan and protonated RCDs are positively charged, being designated as Chit+ and RCD+, respectively; for details, Appendix A. Such protonated monomers of both polymers have been employed in MD simulations to assess their ability to capture Cu(II) ions in aqueous solutions.

#### 2.5.2. MD Simulations

All MD calculations have employed the GROMACS program [29,30]. The GAFF force field [31] from AMBER [32,33,34] was employed to model Chit+ (Appendix A) and RCD+ (Appendix A), while the parameters for Cu(II) were taken from the work of Zhang et al. [35]. To build the topology of Chit+ and RCD+, the corresponding 3D molecular structures were generated with the Avogadro program [36]. The GAMESS package [37] was used to optimise the structures at the RHF/6-31G* theoretical level and then the partial atomic charges were calculated by applying the RESP protocol of the RED program [38]. The geometry and topology input files for GROMACS were generated with the AnteChamber Python Parser interface (ACPYPE) tool [39,40]. For each simulation, either Chit+ or RCD+ was placed in the centre of a cubic box, with the Cu(II) ion being separated from such a molecule by a distance of about 15 nm. These species were then solvated with water molecules described by the TIP4P 2005 potential function [41]. The preparation of the simulation box ended with the addition of three Cl^−^ counterions to neutralise the positive charge of Chit+ (or RCD+) and Cu(II) species.

The whole system in the simulation box was first subjected to energy minimisation to reduce the repulsive interactions resulting from very close-located molecules. Then, the equilibration of temperature and pressure was carried out by running two subsequent simulations of 500 ps, and employing the NVT and NPT ensembles, respectively. It should further be noted that the average temperature was set at 300 K by using the velocity-rescaling thermostat [42,43] with a coupling time of 0.1 ps. Furthermore, the NPT simulation employed the Berendsen barostat [44] to keep the average pressure at 1 bar, with a coupling time of 2 ps.

Ten trajectories of 500 ns were run for each set of initial conditions. Different initial orientations of the Chit+ (or RCD+) species in relation to Cu(II) were explored for the ten trajectories. The integration of the classical equations of motion was carried out with the leapfrog algorithm; a time step of 2 fs was always employed in MD calculations. To impose bond constraints, the LINKS (linear constraint solver) algorithm [45] implemented in GROMACS was applied. The simulations used periodic boundary conditions, with a cut-off value of 10 Å applied for both Coulomb and van der Waals interactions. Long-range electrostatic energy was evaluated using the particle-mesh Ewald method [46,47].

The analysis of MD simulations focused on the following properties. The calculation of dihedral angle distributions for Chit+ (or RCD+) species was carried out over the ten trajectories. Likewise, both the radial and spatial distribution functions (designated RDF and SDF, respectively) of Cu(II) around Chit+ (or RCD+) were calculated as an average over the ten trajectories. Conversely, clustering analysis as well as the representation of the Cu(II)–Chit+ and Cu(II)–RCD+ distances as a function of the time were performed only for a typical trajectory. Only the SDF was calculated with the TRAVIS program [48,49]; all the other analyses employed computational tools available in the GROMACS package [50].

## 3. Results and Discussion

### 3.1. Synthesis and Characterisation of RCDs 

As described in Section 2.3, reduced chitosan derivatives were obtained by a one-pot strategy divided into two steps, whose synthetic sequence is schematised in Figure 1.

The calculated values of *DM* and *DR* for RCD1–4 using the Equations (1) and (2) are depicted in Table 2 and the degree of deacetylation (Equation (3)) obtained for unmodified chitosan was 78%, which is consistent with the supplier’s indications.

It can be concluded from the analysis of Table 2 that the use of MeOH as solvent leads to better yields. However, the ability to adsorb Cu(II) seems to be higher for RCD3 than for RCD4 (see Section 3.2.1 below). Thus, the characterisation was focused on that material. 

The infrared spectra of chitosan, reduced chitosan (RCD3) and salicylaldehyde are shown in Figure 2A. Chitosan and RCD spectra showed a C–O antisymmetric stretching from β-(1→4) glycosidic bonds at 1150 cm^−1^, as well as other characteristic polysaccharide bands at 1059 and 1021 cm^−1^. Additionally, the following vibrational modes can be assigned: C–N axial deformation of amide groups at 1314 cm^−1^; acetyl CH_3_ symmetrical angular deformation at 1375 cm^−1^; C-N axial deformation of amine groups at 1420 cm^−1^; N–H bending vibrations at 1560 cm^−1^; C=O stretching band (amide I) at 1655 cm^−1^ proving the incomplete deacetylation of chitin; C–H stretching at 2871 cm^−1^; and both O–H and N–H stretching overlapped at 3289–3295 cm^−1^, which appeared as a broad band due to hydrogen bonds. The modification of chitosan with salicylaldehyde to form an RCD was confirmed by the peaks at 755 cm^−1^, 1249 cm^−1^ and 1457–1491 cm^−1^, assigned to C–H out of plane bending of the aromatic ring, C–O phenolic stretching and C=C stretching of the aromatic groups, respectively. Since the reduction step was successful, RCD3 shows only a residual peak at 1588 cm^−1^, due to C=N elongation vibrations of the remaining imine bonds [51,52].

The thermal stability of the polymers was evaluated through thermogravimetric analysis (TGA). The thermograms in Figure 2B showed that the chitosan and RCD3 have two significant mass-loss steps: (1) centred at 60 °C and 55 °C for chitosan and RCD3, respectively, due to a desolvation process; and (2) at 292 °C (chitosan) and 282 °C (RCD3), assigned to the polysaccharide pyrolytic chain degradation, in particular the amino bond, which requires the lowest activation energy [53]. The slight decrease in the thermal stability of RCD3 can be justified by the presence of the aromatic ring with the consequent decrease in intermolecular chain interactions. It can also be observed that the overall mass-loss percentage for reduced chitosan was slightly lower compared to commercial chitosan, and the modification step led to the appearance of a small mass loss at around 400 °C for RCDs, suggesting some changes in the substituent fraction of the biopolymer at higher temperatures.

Although differences in *DM* and *DR* values were obtained for the various RCDs, no significant changes were observed in the FTIR-ATR spectra and in the TGA characterisation between the RCDs.

### 3.2. Heavy Metal Adsorption 

#### 3.2.1. Preliminary Studies of Cu(II) Adsorption

Preliminary adsorption studies were performed to identify the RCD with the best Cu(II) adsorption performance. As can be seen from the analysis of Figure 3, RCD3 stands out from the other polymers, having a Cu(II) removal efficiency (66.2%) that surpasses even that of unmodified chitosan (58.7%). Thus, among the synthesised polymers, RCD3, with a moderate percentage of modification (43%) and a higher percentage of reduction (98%), showed the best results for the adsorption of Cu(II) and was chosen for further sorption analysis. The better performance of RCD3 compared to RCD1 and RCD2 is due to its higher percentage of reduction. Furthermore, despite the higher *DM* value, RCD4 showed a lower adsorption efficiency than RCD3 as it floats on water, reducing contact between the active sites of the polymer and the adsorbate in solution. This flotation can be explained by the decrease in the polymer compaction and density with the increase in the degree of modification. The better performance of RCD3 compared to RCD1 and RCD2 is due to its higher percentage of reduction.

#### 3.2.2. Effect of the Solid–Liquid Ratio on Cu(II) Adsorption

The study of the effect of the solid-liquid ratio on the adsorption of Cu(II) onto chitosan and RCD3 was also carried out. From the results shown in Figure 4, it is possible to observe an increasing trend towards adsorption efficiencies as a function of *R*_S/L_ using RCD3, while with chitosan as sorbent, the adsorption capacity remains constant to *R*_S/L_ above 2.5 mg mL^−1^. This effect can be explained by the fact that RCD3 becomes a gel-like substance by swelling in water, unlike chitosan, which improves its surface area, porosity, and dispersity in water, and consequently the adsorption capacity for copper(II) ions binding [23]. In addition to the profile observed in the adsorption efficiencies, a reduction in the polymer adsorption capacity was visualised with the increase in *R*_S/L_ (i.e., with a greater polymer mass). However, from the relationship between *Q* and *q_e_*, it was found that *R*_S/L_ = 2.5 mg mL^−1^ would be the best condition for further adsorption studies.

#### 3.2.3. Effect of pH on Cu(II) Adsorption

To evaluate the role of pH in the Cu(II) adsorption by chitosan and RCD3, the initial pH of metal solutions was varied in the range of 3–6 (Figure 5). It was observed that at pH = 3–5, higher adsorption efficiencies were obtained with RCDs than with neat chitosan (70% vs. 63% at pH = 3; 80% vs. 69% at pH = 4; and 79% vs. 71% at pH = 5, for RCD3 and chitosan, respectively), while similar removal efficiencies were achieved with both polymers at pH = 6 (72% vs. 73%, respectively). Studies at higher pH levels were not possible due to copper hydroxide precipitation [54,55]. Since between pH 4 and 6 there were no significant changes in chitosan adsorption capacity and the best results obtained with RCD3 were at initial pH of 4, the latter value was chosen for the study of the adsorption mechanism. It can be mentioned that these values are of the same order of magnitude as those previously reported for similar conditions [56,57].

#### 3.2.4. Effect of the Initial Concentration on Cu(II) Adsorption

The influence of Cu(II) aqueous solution concentration on metal ion adsorption can be seen in Figure 6A. As shown, adsorption tests were performed with initial concentrations between 10 and 500 mg L^−1^ for modified (RCD3) and unmodified chitosan, and we concluded that RCD3 tends to be slightly more efficient in Cu(II) removal, especially at lower concentrations (25–100 mg L^−1^). Additionally, the decrease in efficiencies with increasing copper concentration is associated with the saturation of active sites (e.g., amine and hydroxyl groups) on the polymers’ surface. In fact, from the analysis of sorption isotherms (Figure 6B), it has been found that the Langmuir and Langmuir–Freundlich models are those that best fit to the experimental data for chitosan and RCD3, respectively (Appendix A).

It was also possible to notice that RCD3 shows a higher maximum Cu(II) adsorption capacity ((78 ± 11) mg g^−1^)) than chitosan ((54 ± 2) mg g^−1^)), as well as other natural and unmodified low-cost adsorbents and chitosan-based polymers [58,59], namely some modified and cross-linked chitosan materials described in the literature [60]. Additionally, in several studies, it was also possible to identify some complex polymers that result from the combination of chitosan with other materials (e.g., polystyrene, graphene oxide, metal–organic structure, or cellulose) [60], as well as other composite materials [61], which possess higher adsorption capacities. However, it should be pointed out that the strategy used in this work allowed us to obtain a very efficient adsorbent only after a simple, easy, and cost-effective chitosan modification.

#### 3.2.5. Cu(II) Adsorption Kinetics

Two important physicochemical aspects for the evaluation of the sorption process as a unit operation are the equilibrium and the kinetics of sorption. As such, Cu(II) uptake at pH = 4 and 25 °C, starting from a 100 mg L^−1^ solution and *R*_S/L_ = 2.5 mg mL^−1^, was also measured as a function of contact time, as shown in Figure 7. The studies of adsorption kinetics using chitosan and RCD3 as adsorbents were carried out between 30 min and 1440 min (24 h). It was concluded that the adsorption rate was quite fast during the first few minutes and started to stabilise after 480 min, when the equilibrium plateau began to form. The adsorption kinetics of Cu(II) by the two polymers were relatively similar and again the greater tendency of RCD3 to adsorb a larger amount of Cu(II) was visualised, although in the initial moments the adsorption by RCD3 seemed to be slower compared to that of chitosan.

To better understand the adsorption process, pseudo-first-order and pseudo-second-order models were fitted to the experimental data [6]. However, as shown in Appendix A, the best representation of the results came from the pseudo-second-order model, according to the goodness-of-fit criteria [62]. This model hints at an adsorption limited by the surface reaction, which is explained by the formation of metal–ligand coordination bonds [6].

#### 3.2.6. Experiments of Ion Metal Selectivity

Selectivity is one important property to consider when developing adsorbent materials; in this particular study, this was the adsorption of Cu(II). To evaluate the selectivity of RCD3 and chitosan, comparisons of the adsorption efficiencies of both polymers for Cu(II), Ni(II), Cd(II) and Pb(II) were made in a mixed component solution, containing each of the metals at a concentration of 0.0004 M. These other divalent metals are also very common and can be found alongside copper in polluted sites [1]. In Figure 8, significant differences in the adsorption efficiencies of each of the competing metals can be observed, leading to the conclusion that both RCD3 and chitosan are highly selective for the adsorption of copper ions in water, even in the presence of other heavy metal ions as interferents.

#### 3.2.7. SEM and SEM-EDS Characterisation

The effect of Cu(II) adsorption on the surface morphology of the adsorbent materials (RDC3 and chitosan) was studied microscopically by SEM analysis, and Figure 9 shows the differences between the polymer morphologies before and after Cu(II) adsorption. By analysing Figure 9A,B it can be seen that prior to Cu(II) adsorption, both RCD3 and chitosan exhibit a granular structure. However, neat RCD3 shows a more heterogeneous surface morphology and size compared to pristine chitosan. The less compacted granules with a rougher surface of RCD3 can explain the better interaction with Cu(II) and consequently its better adsorption capacity [63]. After Cu(II) adsorption from a 100 mg L^−1^ solution at pH = 4 (Figure 9C,D), Cu(II)-loaded RDC3 shows a more porous and rougher surface, and large aggregates compared to Chitosan+Cu(II), which may be due to the higher adsorption efficiency of RCD3. Metalated polymers show a more compact structure compared to non-modified ones, possibly because the copper ions form a bridge between the polymer chains by coordination reaction. 

By SEM-EDS, a homogeneous dispersion of the constituent elements of the polymers (e.g., carbon, oxygen and nitrogen) and Cu(II) was observed, which means a regular distribution of the metal ions over both polymer surfaces during Cu(II) adsorption. The copper weight percentage of around 3% that was observed in the EDS spectra is further evidence of the effectiveness of the sorption process (Appendix A). The EDS spectra of the polymers recovered after the selectivity tests show a greater selectivity for Cu(II) adsorption, as this was almost always the only metallic species that was observed in different sites on the surface of both RCD3 and chitosan after adsorption (Appendix A, respectively). However, in some regions, peaks attributable to nickel were also observed, albeit in insignificant amounts, which can confirm the trends shown in Figure 8. No peaks corresponding to cadmium and lead were observed.

### 3.3. MD Results

The experimental results show that the functionalisation of chitosan favours the capture of Cu(II) in water. MD simulations have been employed to obtain molecular-level insight that can be used to rationalise this trend. Accordingly, a set of MD calculations have been performed for both Chit+ and RCD+ in an aqueous solution of Cu(II). Figure 10 displays the distances between Cu(II) and the oxygen atoms of Chit+ for a typical trajectory; also shown by the black lines in this figure are the corresponding distances for a typical trajectory with Chit+. It is apparent from Figure 10 that Cu(II) can bind both Chit+ and RCD+, but the Cu(II)–RCD+ complex tends to form more rapidly. Although this is a general trend, there are trajectories in which neither Cu(II)–Chit+ nor Cu(II)–RCD+ complexes occur in 500 ns. It is particularly interesting to note in Figure 10 that the distances between the oxygen atoms of Chit+ (or RCD+) and Cu(II) decrease significantly when the complex is formed. It may be observed that the corresponding smallest distances are those involving the oxygen of the ring and the neighbouring hydroxyl groups, regardless of considering Chit+ or RCD+. In turn, the distances between the Cu(II) and the remaining hydroxyl groups are larger and oscillate much more, which seems to indicate that such groups do not directly participate in the formation of the complex. Accordingly, the corresponding RDFs represented in Figure 11 show the highest narrow peaks associated with the smallest Cu(II)–oxygen distances that are directly involved in the formation of the complex (Figure 11D–F).

Additionally, the oscillatory behaviour of the larger distances is also apparent in the broader peaks of the corresponding RDF functions. This is an indication of a great variation in the geometry of such groups, and it is particularly significant for the hydroxyl of the phenol group (Figure 11A). Indeed, the cluster analysis of the RCD+–Cu(II) complex shows three main structural motifs for RCD+ in Figure 12 that correspond to the variation of the position of the *o*-methylene-phenol group in relation to the chitosan ring. Whereas the central RCD+ motif in Figure 12 is essentially planar, the other two structures show orientations of the benzene ring either up or down to the chitosan ring. Inspecting the corresponding trajectories shows that the up and down configurations, somehow forming a cage over the chitosan ring, appear to aid in the stabilisation of the RCD+–Cu(II) complex. This effect is apparent in the SDFs of Appendix A. It is worth noting in this figure that the maximum of probability of finding Cu(II) is localised over the abovementioned three oxygens of chitosan for Chit+, while it spreads over the whole ring in the case of RCD+, thus showing an assistance role for the stabilisation of the RCD+Cu(II) complex by the *o*-methylene-phenol group. 

Moreover, we have also investigated the flexibility character of the structure of both Chit+ and RCD+, since it can explain in some way the formation of configurations that favour (or disfavour) the complex formation. The distribution of some representative dihedral angles is shown in Appendix A. Regarding the dihedral angles associated with the chitosan ring, there are no significant differences between Chit+ and RCD+. The only exceptions arise for the dihedral angles associated with the *o*-methylene-phenol group (Appendix A). In these figures, additional peaks arise for RCD+ at −145 and 155 degrees, respectively. The preference for these specific dihedral angles for RCD+ reveals the importance of the assistance given by the *o*-methylene-phenol group for the formation of the RCD+Cu(II) complex.

Regarding the dihedral angles associated with the substituent group of RCD+, it may be observed in Appendix A that such group shows a great mobility, i.e., it explores several structural motifs (e.g., a cage and a chair-type structure in different angles) that may aid in the capture of Cu(II).

## 4. Conclusions

To promote pollutant removal from aqueous media, in this work, reduced chitosan derivatives (RCDs) were prepared by a one-pot two-step synthesis involving the modification of chitosan with salicylaldehyde and the subsequent reduction of the formed imine bonds. This straightforward procedure turns a polymer that can be obtained from fishery waste into a value-added product. Changes in the chitosan structure were confirmed by FTIR-ATR and also by ^1^H-NMR analysis, through which it was observed that the modification and reduction percentages were influenced by the solvent. Furthermore, the thermal stability of chitosan did not show significant changes after derivatisation.

Among the RCDs, RCD3, with a 43% degree of modification and a 98% degree of reduction, was the most efficient derivative for Cu(II) removal from water, also showing a higher adsorption capacity than pristine chitosan. Adsorption studies confirmed that the best conditions were at pH = 4 and *R*_S/L_ = 2.5 mg mL^−1^, and that, under these conditions, RCD3 proved to be a better Cu(II) adsorbent than chitosan, mainly at low concentrations. It should be stressed that, due to functionalisation, the number of active adsorption sites in RCD3 is lower than in chitosan; even so, the removal efficiency is higher. This clearly shows the outcome of our strategy on the adsorption process. The rougher surface of RCD3 observed by SEM may explain the higher Cu(II) affinity, and the more compact structure of Cu(II)-loaded polymers can be attributed to the copper ability to interconnect polymeric chains by coordination reaction. By analysing the sorption isotherms, Langmuir and Langmuir–Freundlich models were the ones that best fit the experimental data for chitosan and RCD3, respectively, and the pseudo-second-order kinetic model best described the kinetic data, confirming the occurrence of chemisorption, explained by the formation of metal–ligand coordination bonds. Both RCD3 and chitosan also showed higher selectivity for Cu(II) adsorption than Ni(II), Cd(II) and Pb(II). 

Molecular-level insight into Cu(II)–polymer interactions obtained by MD simulations revealed the importance of the mobility of the *o*-methylene-phenol group in RCD3 for the greater capture of Cu(II) and the stability of the final complex, and also showed that Cu(II) interacts more favourably with the oxygen atom of the glucosamine ring and the neighbouring hydroxyl groups, both in RCD3 and chitosan. The stabilisation of metal ions occurs as a consequence of a scorpion-tailed structure. On the basis of all experimental data, RCD3 is an efficient adsorbent that can be used for Cu(II) removal from wastewater, also being greater compared to several materials described. Thus, the new, simple, and effective strategy described shows great potential for using highly available and non-toxic biopolymers as a starting material, to improve its properties, efficacy, and selectivity as well as its application in adsorption processes. 

## Figures and Tables

**Figure 1 molecules-28-04110-f001:**
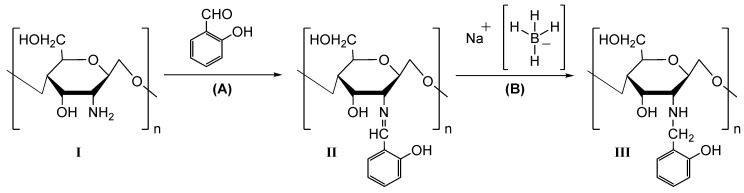
Synthetic approach for the preparation of reduced chitosan derivatives (RCDs).

**Figure 2 molecules-28-04110-f002:**
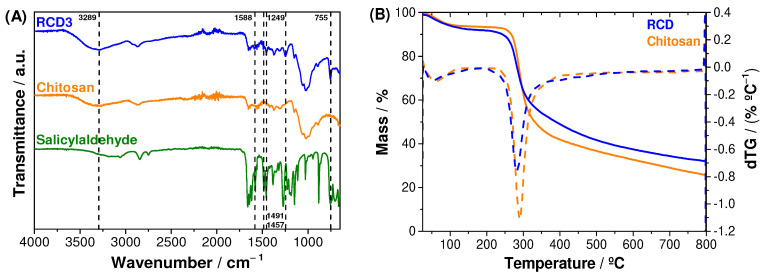
(**A**) FTIR-ATR spectra of salicylaldehyde, chitosan and RCD3; (**B**) Thermograms (solid line) and respective dTGs (dotted line) for chitosan and RCD3.

**Figure 3 molecules-28-04110-f003:**
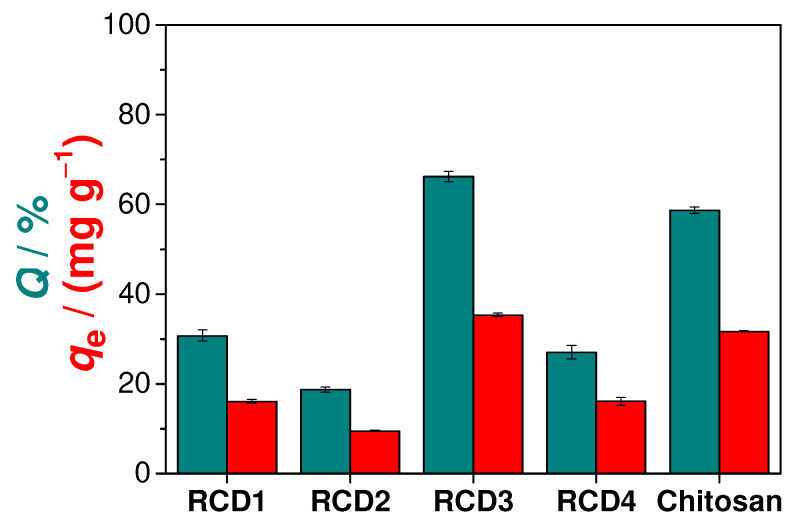
Cu(II) adsorption efficiencies (*Q*, green bars) and capacities (*q_e_*, red bars) of different polymers (RCDs and chitosan).

**Figure 4 molecules-28-04110-f004:**
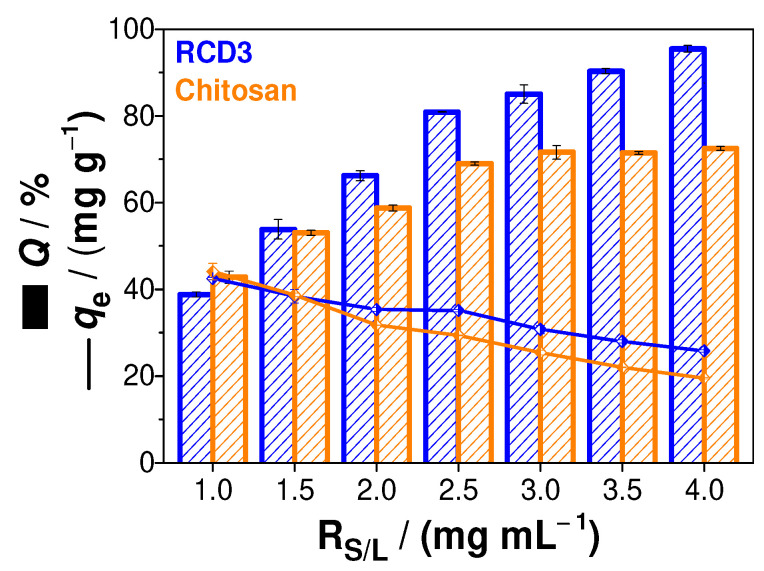
Sorption efficiency (*Q*, %) and the amount of metal ion sorbed per gram of adsorbent (*q_e_*, mg g^−1^) as function of solid-liquid ratio (*R*_S/L_, mg mL^−1^). Experimental data for RCD3 and chitosan are in blue and orange, respectively.

**Figure 5 molecules-28-04110-f005:**
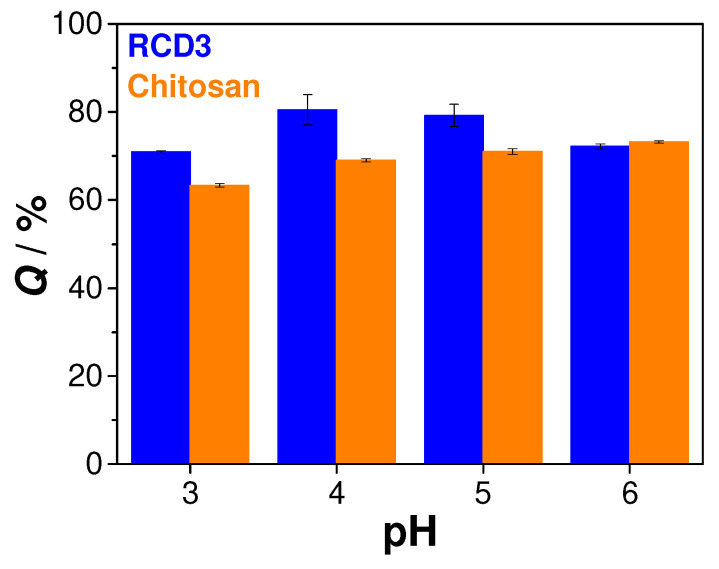
Sorption efficiency of RCD3 and chitosan as function of pH (*C*_0_ = 100 mg L^−1^ and *R*_S/L_ = 2.5 mg mL^−1^). Experimental data for RCD3 and chitosan are in blue and orange, respectively.

**Figure 6 molecules-28-04110-f006:**
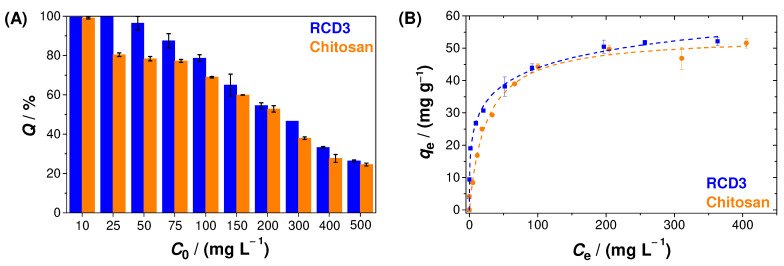
(**A**) Effect of Cu(II) initial concentration on copper adsorption efficiency; (**B**) Cu(II) adsorption isotherms, at 25 °C, by RCD3 and chitosan. The dashed lines represent the fitting of different models to the experimental data: Langmuir (chitosan) and Langmuir–Freundlich (RCD3). Experimental data for RCD3 and chitosan are in blue and orange, respectively.

**Figure 7 molecules-28-04110-f007:**
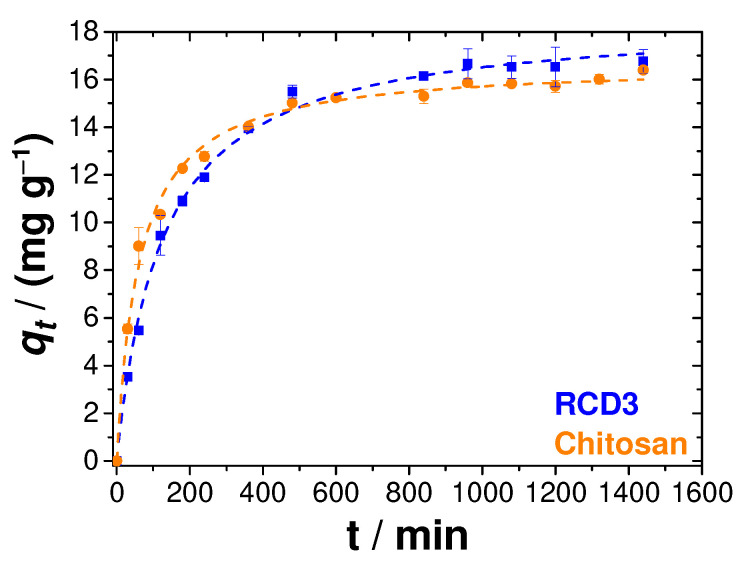
Cu(II) adsorption kinetics, at 25 °C, by RCD3 and chitosan, starting from a metal solution of 100 mg L^−1^. The dashed lines represent the fitting of the pseudo-second-order model to the experimental data. Experimental data for RCD3 and chitosan are in blue and orange, respectively.

**Figure 8 molecules-28-04110-f008:**
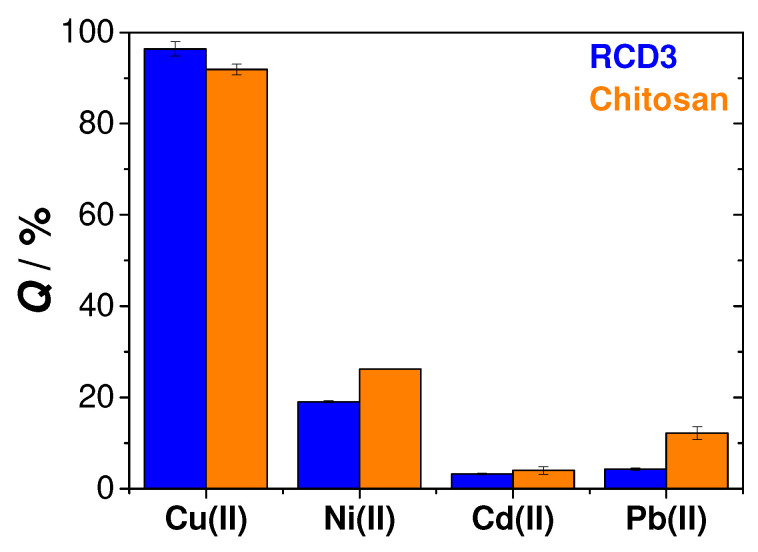
Selectivity tests for the adsorption of Cu(II) ions by RCD3 and chitosan. Experimental data for RCD3 and chitosan are in blue and orange, respectively.

**Figure 9 molecules-28-04110-f009:**
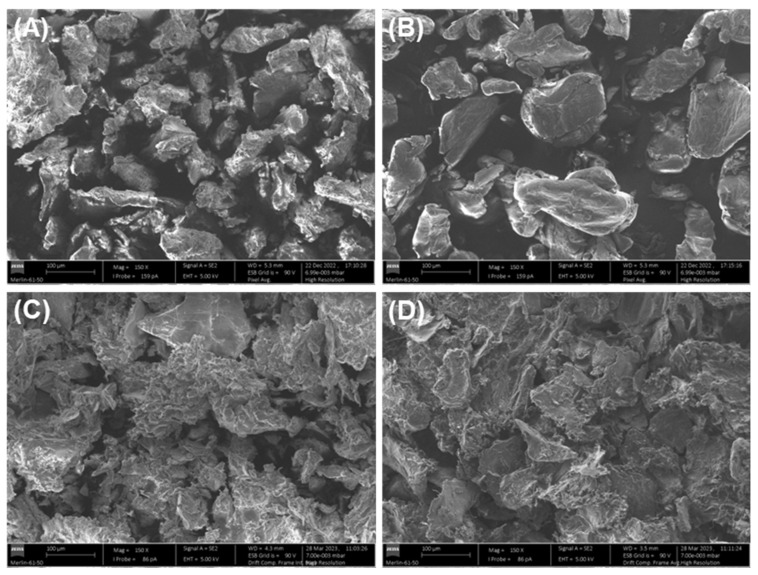
SEM micrographs of neat (**A**) RCD3 and (**B**) Chitosan; and after Cu(II) adsorption (**C**) RCD3+Cu(II) and (**D**) Chitosan+Cu(II). Magnification 150×.

**Figure 10 molecules-28-04110-f010:**
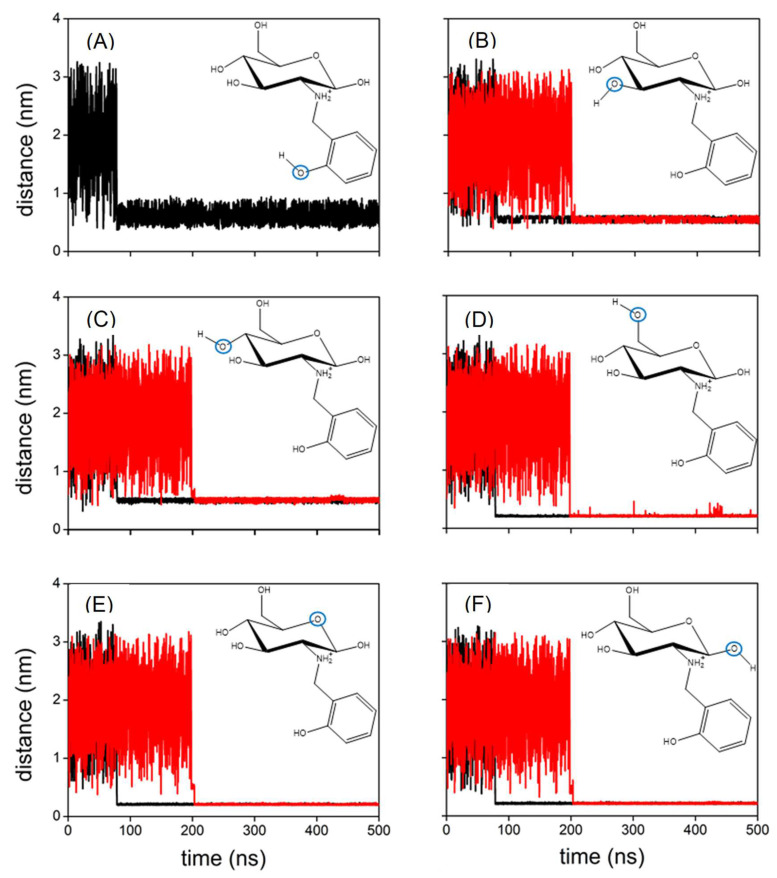
(**A**) Distance between Cu(II) and the indicated oxygen atom (blue circle) of RCD+ (black line); (**B**–**F**) distance between Cu(II) and the indicated oxygen atom (blue circle) of RCD+ (black lines) and Chit+ (red lines), respectively; all panels represent a typical trajectory.

**Figure 11 molecules-28-04110-f011:**
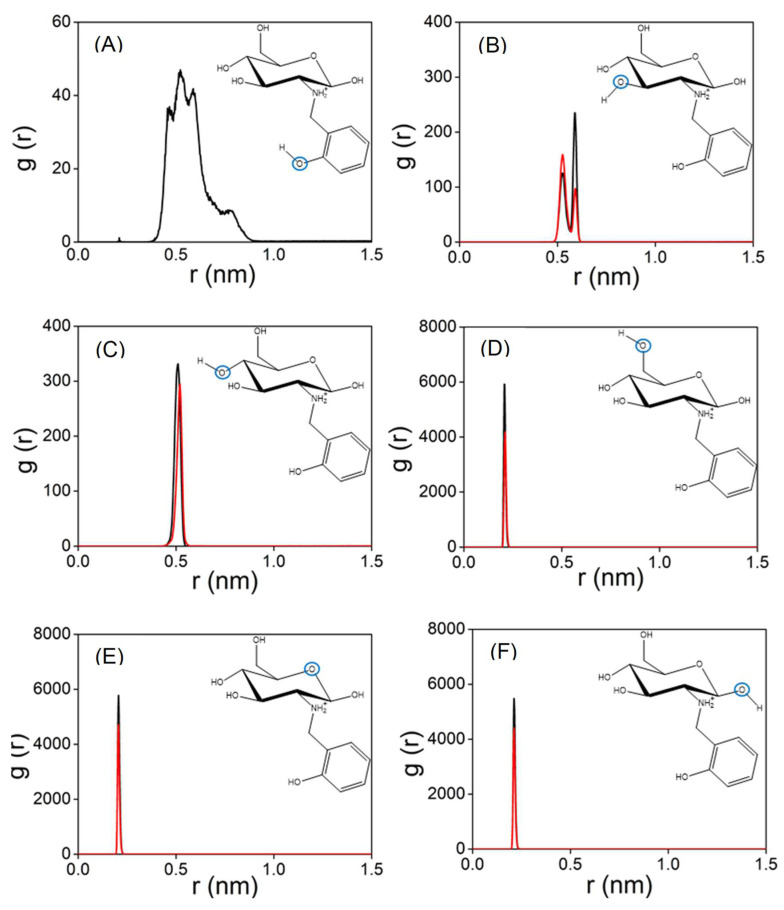
(**A**) RDF of RCD+–Cu(II) (black line); (**B**–**F**) RCD+–Cu(II) (black line) and Chit+–Cu(II) (red lines), repectively. In each panel, the RDF is calculated for the distance between Cu(II) and the indicated oxygen atom (blue circle).

**Figure 12 molecules-28-04110-f012:**
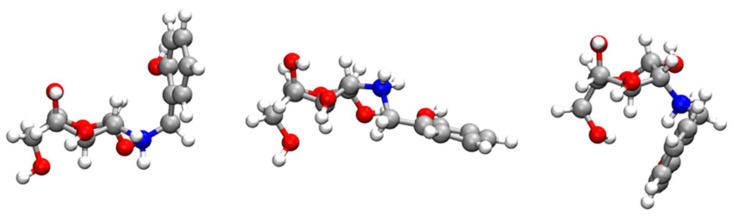
Main RCD+ structures arising from the clustering analysis carried out over the period of a typical trajectory where the RCD+–Cu(II) complex is formed. From left to right, the probability of having a particular RCD+ structure is 57%, 22% and 5%, respectively. The atoms are represented by the following colours: oxygen (red), carbon (silver), nitrogen (blue) and hydrogen (white).

**Table 1 molecules-28-04110-t001:** Solvents used in the synthesis of chitosan derivatives.

Chitosan Derivative	Step (A)	Step (B)
RCD1	CH_3_COOH (2%)	CH_3_COOH (2%):MeOH (1:1, *v*/*v*)
RCD2	CH_3_COOH (2%):MeOH (1:1, *v*/*v*)	CH_3_COOH (2%)
RCD3	CH_3_COOH (2%):MeOH (1:1, *v*/*v*)	MeOH
RCD4	MeOH	MeOH

**Table 2 molecules-28-04110-t002:** Degrees of modification (*DM*) and reduction (*DR*) for chitosan derivatives.

Chitosan Derivative	*DM*/%	*DR*/%
RCD1	42	69
RCD2	43	63
RCD3	43	98
RCD4	69	98

## Data Availability

Not applicable.

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
