# Peer review of "Reduced Chitosan as a Strategy for Removing Copper Ions from Water"

_molecules, 2023, doi:10.3390/molecules28104110_

Round 1

Reviewer 1 Report

The article "Reduced chitosan as strategy for removal metal ions from water” describes a simple selective method of removing toxic copper ions from water with modified chitosan. The authors have done a good job but need to be improved per the comments below.

1. In the introduction, why was salicylic aldehyde chosen as a modifying agent to chitosan and why was the imine bond reduced? Due to the hydrolysis of Schiff bases in aqueous solutions?

2. An explanation is required for why the more modified chitosan with a salicylic aldehyde (RCD4) is less effective than RCD3 (Figure 3). There are more copper ion binding sites in RCD4.

3. In the selectivity test (Figure 4 F), it would be interesting to look at Fe3+ ions. They are often found in natural waters and have a large charge. Therefore, they can act as interfering ions in removing copper ions from the water.

Reviewer 3 Report

The paper "Reduced chitosan as strategy for removal metal ions from water" can be accepted for publication, but several areas need improvement. I recommend this manuscript for publication after the clarification of the following major comments

1) The English should really be improved and revised. There are a lot of sentences with no sense and a number of grammatical and typographical errors. Extensive editing of English language and style required.

2)  Abstract: Need to revise. In the abstract, add a description to clarify the flow of work. Authors are advised to revise the abstract and please focus the abstract on your review approach.

3) The introduction part needs to be strengthened. Authors must elaborate introduction part by indicating the novelty of the study and emphasizing the role of different adsorbents explored recently for the removal of the metal ions from their aqueous solutions. I am suggesting here some recent important research papers on the subject matter, which authors are suggested to cite in the revised manuscript:

-https://doi.org/10.1016/j.envres.2023.115606

-https://doi.org/10.1007/s42250-022-00350-3

-https://doi.org/10.1016/j.chemosphere.2023.137922

4) Why do the authors study the adsorption of Cu(II) onto reduced chitosan?

5) Why was reduced chitosan chosen as an adsorbent in this study? What is the chemistry behind the removal using this adsorbent?

6) The authors' interpretation and discussion of the results should be improved.

7) Have you compared the results obtained using other adsorbents for example?

8) The conclusion also needs to be rewritten. Include the following: new concepts and innovations demonstrated in this study; a summary of findings; a comparison with findings by other workers; and a concluding remark. 

need improve

Round 2

Reviewer 3 Report

Accept in present form